# Genome-Wide Association Mapping Identifies New Candidate Genes for Cold Stress and Chilling Acclimation at Seedling Stage in Rice (*Oryza sativa* L.)

**DOI:** 10.3390/ijms232113208

**Published:** 2022-10-30

**Authors:** Jianguo Li, Ahmed Adel Khatab, Lihua Hu, Liyan Zhao, Jiangyi Yang, Lingqiang Wang, Guosheng Xie

**Affiliations:** 1State Key Laboratory for Conservation & Utilization of Subtropical Agro-Bioresources, College of Agriculture, Guangxi University, Nanning 530004, China; 2College of Plant Science & Technology, Huazhong Agricultural University, Wuhan 430070, China; 3College of Life Science & Technology, Guangxi University, Nanning 530004, China

**Keywords:** rice, fresh weight, chilling acclimation, cold stress, GWAS, RNA-seq

## Abstract

Rice (*Oryza sativa* L.) is a chilling-sensitive staple food crop, and thus, low temperature significantly affects rice growth and yield. Many studies have focused on the cold shock of rice although chilling acclimation is more likely to happen in the field. In this paper, a genome-wide association study (GWAS) was used to identify the genes that participated in cold stress and chilling accumulation. A total of 235 significantly associated single-nucleotide polymorphisms (SNPs) were identified. Among them, we detected 120 and 88 SNPs for the relative shoot fresh weight under cold stress and chilling acclimation, respectively. Furthermore, 11 and 12 quantitative trait loci (QTLs) were identified for cold stress and chilling acclimation, respectively, by integrating the co-localized SNPs. Interestingly, we identified 10 and 15 candidate genes in 11 and 12 QTLs involved in cold stress and chilling acclimation, respectively, and two new candidate genes (*LOC_Os01g62410*, *LOC_Os12g24490*) were obviously up-regulated under chilling acclimation. Furthermore, *OsMYB3R-2* (*LOC_Os01g62410*) that encodes a R1R2R3 MYB gene was associated with cold tolerance, while a new C3HC4-type zinc finger protein-encoding gene *LOC_Os12g24490* was found to function as a putative E3 ubiquitin-protein ligase in rice. Moreover, haplotype, distribution, and Wright’s fixation index (FST) of both genes showed that haplotype 3 of *LOC_Os12g24490* is more stable in chilling acclimation, and the SNP (A > T) showed a difference in latitudinal distribution. FST analysis of SNPs in *OsMYB3R-2* (*LOC_Os01g62410*) and *LOC_Os12g24490* indicated that several SNPs were under selection in rice *indica* and *japonica* subspecies. This study provided new candidate genes in genetic improvement of chilling acclimation response in rice.

## 1. Introduction

Rice (*Oryza sativa*) is one of the three major cereal crops used as a staple food by more than half of the world population [1]. As rice originated in tropical or subtropical areas, it is generally sensitive to cold stress [2,3]. Low-temperature stress is the main limiting factor of rice cultivation, as it affects crop yield and quality. Therefore, it is important to develop more robust and cold-stress-resilient rice germplasms. The identification of genes involved in cold tolerance is a crucial aspect of rice breeding. 

Plants have evolved sophisticated regulatory networks to cope with temperature environments. Asian cultivated rice is represented by two subspecies, *indica* and *japonica*, and each of them includes many accessions exhibiting different tolerance levels to cold stress [4]; the underlying genetic variation has become a powerful tool for the discovery of many genetic loci and cold-related genes in rice using these genetic resources [5]. It is interesting to find out the physiological and molecular mechanisms underlying their differences in plant responses to cold stresses, which can provide new power for engineering cold-tolerant and high-yielding rice varieties.

In addition, there are two kinds of low temperature in rice: (1) rapid decline of the temperature and (2) chilling stress, during which the temperature gradually declines to an unfavorable range for rice growth. Most studies of cold-stress responses in rice are close to the rapid decline of the temperature even though chilling acclimation is more likely to happen in rice growth and development. Therefore, it is necessary to study the mechanisms underlying chilling acclimation and find the candidate genes for the genetic improvement of rice cold tolerance.

In recent years, genome-wide association study (GWAS) has become an important method for identifying QTL and genomic regions in many species [6,7,8]. For example, Liu performed a GWAS of the tiller response to nitrogen that is most closely correlated with nitrogen-use efficiency in rice and identified a candidate gene *OsTCP19* [9]. Overall, 51 QTLs were detected by genome-wide association study by using 174 Chinese rice accessions for cold tolerance [10]; 132 QTLs were detected in 527 rice cultivars for both rice natural chilling and cold shock stresses [11]; 56 novel loci were detected for cold stress at seedling stage [12], and a similar analysis was carried out by using a rice population of 2262 [13]. In recent studies, RNA-seq has been widely used in uncovering the causal genes in GWAS in plants [14,15].

To gain insight into the mechanisms that have enabled rice to endure cold environments, we performed a GWAS to elucidate the cold tolerance in rice. We collected 338 accessions of rice core germplasms and evaluated the performance of the rice seedling under two kinds of cold stress, i.e., cold shock and cold acclimation, by detecting the relative changes of shoot fresh weight, which is an ideal symbol for cold stress adopted by previous studies [16,17,18,19]. Fixed and random model circulating probability unification (FarmCPU) was also used in this study, which is one of the multi-locus models and is more robust in controlling both false positives and negatives [20,21]. In addition, RNA-seq data, haplotype, and distribution analysis were finally carried out for dissecting causal genes in GAWS. Thus, our study provides a further understanding of the genetic regulation for cold-stress tolerance in rice.

## 2. Results

### 2.1. Different Responses of Indica and Japonica Rice Seedlings and Trait Correlations under Different Temperature Conditions

In this study, shoot fresh weight was adopted to evaluate the cold response of the seedlings of 338 natural rice accessions under different temperature conditions (Appendix A). Firstly, statistics of three traits were presented, including the mean, standard deviation (SD), minimum (min), maximum (max), coefficient of variation (CV), skewness, and kurtosis analyses, which were all carried out (Appendix A). The variation of the shoot fresh weight under normal temperature (SWNT) is relatively small. However, upon low-temperature treatments, a larger variation was observed, as indicated by the CV of the trait (Appendix A). Then, 338 rice accessions were allocated to three subpopulations (*indica*, *japonica,* and *intermediate*) according to previous study [22]. It was shown that the shoot fresh weight of *japonica* accessions was significantly higher than that of the *indica* accessions under normal temperature (Figure 1A). After low-temperature treatments, the relative shoot fresh weight of *japonica* was significantly lower than *indica* and *intermediate* accessions, suggesting that *japonica* accessions are more tolerant to cold stress than *indica* and *intermediate* accessions (Figure 1B,C). Correlation analysis indicated that relative shoot fresh weight after cold stress (RSWCS) and relative shoot fresh weight after chilling acclimation (RSWCA) are significantly and positively associated. Shoot fresh weight under normal temperature (SWNT) is not associated with that after cold stress and after chilling acclimation exposure (Appendix A).

### 2.2. Population Genetic Analyses and Polymorphic SNPs

In this study, we collected 338 accessions consisting of 167 *indica* accessions, 48 *intermediate* accessions, and 123 *japonica* accessions (Figure 2A). Population structure analysis, including principal component analysis (PCA) and kinship analysis, was performed and showed that two components or clusters separate this panel (Figure 2B,C). Furthermore, the LD of this panel was estimated as 57 kb when r2 dropped to half of the maximum value (Figure 2D) [23]. The genotypes of the 338 accessions were downloaded from the website https://snp-seek.irri.org, accessed on 22 June 2022 [24]. Then, about 3.4 Mb SNPs in the whole rice genome were filtered. In detail, chromosome 4 has the smallest marker density, with one SNP per 103 bp, while the largest marker density was observed on chromosome 10 (1 SNP/99 bp). The number of SNPs and marker density show little difference in 12 rice chromosomes. The average density of SNPs was one SNP per 111 bp (Appendix A).

### 2.3. GWAS of Shoot Fresh Weight under Three Diffferent Conditions

Shoot fresh weight is an ideal symbol for detecting the relative changes of cold stress [16,17,18,19]. To obtain more reliable SNPs, FarmCPU was performed with the first two principal components and kinship by rMVP package [26]. Firstly, the −log(p) was set at 5 to identify significant association signals. Then, 235 SNPs were found for shoot fresh weight under normal temperature, cold stress, and chilling acclimation. In detail, 27 SNPs were identified for shoot fresh weight under normal temperature (Figure 3A and Appendix A), 120 SNPs were identified for relative shoot fresh weight under cold stress (Figure 3B and Appendix A), and 88 SNPs were identified for the relative shoot fresh weight under chilling acclimation (Figure 3C and Appendix A).

In addition, several significant SNPs located at the previously reported genes were annotated (Figure 3A–C). For example, *OsHOX12* (*LOC_Os03g10210*) and *OsLGG* (*LOC_Os11g41890*) were detected for relative shoot fresh weight under cold stress, and both of them have been identified to be associated with panicle development [27,28]. As for the relative shoot fresh weight under chilling acclimation, *OsMYB3R-2* (*LOC_Os01g62410*) was identified to be associated with cold tolerance [29], *OsbZIP23* (*LOC_Os02g52780*) was involved in drought and salt tolerance [30], *OsJMJ704* (*LOC_Os05g23670*) was a disease-resistance regulator [31], *OsRBG1* (*LOC_Os11g30430*) enhanced tolerance to heat and osmatic and salt stress [32], and *Ospita2* (*LOC_Os12g18729*) was also associated with disease resistance [33].

### 2.4. Comprehensive Analysis of Significant SNPs under Cold Stress and Chilling Acclimation Conditions

To obtain reliable results, regions with two or more significant SNPs were selected as QTL [34,35]. As a result, 11 QTLs and 12 QTLs were found for cold stress and chilling acclimation, respectively (Appendix A). For more details, 254 and 220 genes were found in 11 QTLs and 12 QTLs, respectively. Then, these genes were integrated with the RNA-seq data, and 10 and 15 genes of them were expressed in our previous RNA-seq data (Appendix A).

To filter the specific genes associated with chilling acclimation, a comprehensive analysis was carried out. Firstly, we compared SNPs between cold stress and chilling acclimation, and four overlaps were detected and shown in black triangles, suggesting that cold stress and chilling acclimation share several pathways, but most of them are different. In addition, two overlaps were found between normal temperature and chilling acclimation, indicating that these SNPs are not specific to chilling acclimation (Figure 4A). We can also find that 12 QTLs for chilling acclimation labeled with the black star are different from QTLs for cold stress (Figure 4B). Among fifteen genes in twelve QTLs, three genes have relatively higher expressions under chilling acclimation, and two up-regulated genes were selected for further study (Figure 4C). In addition, qRT-PCR was performed for the two candidate genes *LOC_Os01g62410* and *LOC_Os12g24490*. As a result, both of them showed higher expression levels at cold stress (6 °C) for 12 h and 24 h, and *LOC_Os12g24490* was significantly up-regulated at 6 °C for 12h (Figure 4D). 

### 2.5. LD Block, Haplotype, and Distribution Analyses of the Candidate Genes

LD block, haplotype, and distribution analyses were carried out for three selected high-confidence genes. The frequency of haplotypes in each subpopulation was checked, and only the haplotypes with more than twenty accessions were subjected to further analysis.

LD block analysis showed that SNPs within the gene *OsMYB3R-2* (*LOC_Os01g62410*) have a higher LD (Figure 5A). The accessions harboring Hap_2 (mostly *japonica* type) has better performance under both cold stress and chilling acclimation than the others (Figure 5B). There were specific 14 SNPs in Hap_2, showing great differences between *indica* and *japonica accessions* (Figure 5C).

A haplotype network analysis indicated that Hap_1 consists of 112 *indica* and 21 intermediate accessions, Hap_3 consists of all *indica* accessions, while Hap_2 consists of 105 *japonica* accessions, three *indica* accessions, and five *intermediate* accessions. Mor4eover, Hap_4 consists of 13 *intermediate* accessions (Figure 5D). FST analysis showed that several SNPs in Hap_2 have a higher FST value between *indica* and *japonica* subpopulations, which indicated that these SNPs are under selection (Figure 5E).

As noted above, the gene *LOC_Os12g24490* encodes a C3HC4-type domain containing protein. LD analysis showed that there is a higher linkage disequilibrium within the *LOC_Os12g24490* (Figure 6A). Haplotype analysis showed that Hap_3 had the smallest relative shoot fresh weight under cold stress and chilling acclimation (Figure 6B). As expected, there was a specific SNP that occurred in Hap_3 at chr12:13981997 (Figure 6C). Distribution analysis was carried out for 3024 accessions, and the SNP information was downloaded from https://s3.amazonaws.com/3kricegenome/snpseek-dl/ accessed on 22 June 2022. Distribution analysis showed that this SNP is different with latitude (Figure 6D). Then, the haplotype network was performed by a minimum spanning network using PopART (Figure 6E).

To make an informative network, Hap_7 with four accessions was also used, and we found that Hap_7 differed from Hap_3 at chr12:13981997. Furthermore, the mutation between Hap_3 and Hap_7 might bring about a change in response to cold stress. FST analysis of the gene *LOC_Os12g24490* showed that chr12:13981997 has a higher FST (*ind*/*jap*), suggesting that chr12:13981997 is under selection (Figure 6F).

## 3. Discussion

Cold stress and chilling acclimation are limiting factors for rice growth, development, and yield, and previous studies have focused on finding the cold-tolerance genes [36,37,38]. However, few studies have been performed for chilling acclimation, which more likely happens in natural environment.

In this study, we performed a comparative GWAS for cold stress and chilling acclimation. We found that most of the QTLs for cold stress and chilling acclimation are different, while there are also shared QTLs. Notably, we identified two genes, *OsMYB3R-2* (*LOC_Os01g62410*) and *LOC_Os12g24490*, for chilling acclimation. Interestingly, *OsMYB3R-2* could up-regulate the cell division genes such as *OsCycB1;1* (*LOC_Os01g59120*), *OsCycB2;1* (*LOC_Os04g47580*), *OsCycB2;2* (*LOC_Os06g51110*), and *OsCDC20.1 (LOC_Os02g47180*) and enhance cold tolerance [29]. Therefore, *OsMYB3R-2* (*LOC_Os01g62410*) may participate in both cold stress and chilling acclimation. Haplotype (network) analysis indicated that both *OsMYB3R-2* (*LOC_Os01g62410*) and *LOC_Os12g24490* diverged in *indica* and *japonica*. For example, Hap_2 of *OsMYB3R-2* (*LOC_Os01g62410*) and Hap_3 of *LOC_Os12g24490* mainly consisted of *japonica* accessions. According to FST analysis, we discovered that specific SNPs in Hap_2 of *LOC_Os01g62410* had a higher FST value, indicating that these SNPs and haplotypes are under selection.

In our study, there are some QTLs overlapped with those identified in some previous studies, and several genes in QTLs have been verified to participate in cold stress or other abiotic stress. For example, *qrswca5* overlapped with the QTL (chr4: 29737978- 33358474) that contains the cold-tolerance gene *OsAOX1a* [39]. *LOC_Os06g33710* in *qrwcs7* encodes a distinctive class of spermidine synthase involved in chilling response in rice [40]. The *LOC_Os06g37450* in *qrswca6*, a GATA transcription factor, confers cold tolerance by repressing *OsWRYK45-1* at the seedling stage in rice [41]. Interestingly, several QTLs overlapped with drought-tolerance QTLs, such as *qrswca8* and *qrswcs9* that overlapped with *yld11.1* for yield per plant [42]; *qrswca9* that overlapped with *rv12-2* for root volume [43]; *qrswca10* that overlapped with *qtl12.1* for flowering delay; and *qrswca1* that overlapped with *qtl12.1* for drought-response index [44].

In recent years, GWAS has been widely used for identifying QTL and genomic regions of complex quantitative traits. As a routine, there were three important components: genotype, phenotype, and the model in GWAS. As for genotyping, the LD of rice is usually considered as 100 kb, which indicates that at least 4300 SNPs were required to cover the whole genome of rice (430 Mb) for GWAS. However, the increased number of SNPs will achieve higher accuracy of the QTLs mapping in GWAS. In this study, 3.4 M SNPs (average density is 111 bp/SNP) were used, which is efficient for GWAS mapping. Secondly, traits need to be considered clearly to obtain effective results of GWAS. In this study, the relative values were calculated to emphasize the changes in the seedling weight before and after the low-temperature treatments, which ensure that mapped QTLs were specifically involved in stress responses. Finally, a multi-locus model (FarmCPU) adopted by this study was based on multiple SNPs analysis, which could reduce false-positive sites by considering the linkage disequilibrium.

Finding causal variants and the candidate genes is challenging for GWAS since there are so many factors, such as linkage disequilibrium, false positive of the model, and phenotype error, which impact the ability to obtain precise results. RNA-seq analysis, haplotype analysis, and gene annotation could be used to partly solve the problems. For example, the gene *LOC_Os01g62410* a R1R2R3 MYB gene have been studied and enhance cold tolerance by binding to the mitosis-specific activator cis-element in the promotor of *OsCycB1;1*, a G2/M phase-specific gene, and activating its expression [45]. In this study, we also integrated GWAS with the gene expression profiling during low-temperature stresses. The gene *LOC_Os12g24490* is expressed in chilling acclimation, and there is phenotypic variation between different haplotypes and the gene *LOC_Os12g22490*, encoding C3HC4-type RING zinc finger protein function as a putative E3 ubiquitin-protein ligase, which is essential in the regulation of response to abiotic stress, and the cold-tolerance gene in QTL *ctb1* could interact with a subunit of the E3 ubiquitin ligase, SKP1, suggesting that a ubiquitin-proteasome pathway is involved in cold tolerance [45]. Previous studies also found that diverse environmental stresses (including cold stress) induced the expression of *BrRZFP1* (C3HC4-type RING zinc finger protein) in *Brassica rapa*, and overexpression of *BrRZFP1* conferred increased tolerance to cold, salt, and dehydration stresses [46]. Ectopic overexpression of a novel wheat zinc finger transcription factor *TaZnF* conferred heat-stress tolerance and cold and oxidative stresses in *Arabidopsis* [47]. In addition, a gene encoding C3HC4-type RING finger was linked to fruit quality and chilling injury in peach [48]. These results clearly confirmed the roles of candidate genes obtained in our study in the chilling acclimation in rice, and meanwhile, these findings imply the reliability of our GWAS results. It will be of importance to identify the mechanism of both genes in chilling acclimation in crops.

## 4. Materials and Methods

### 4.1. Phenotyping for GWAS

The 338 rice accessions analyzed were from the 3K Rice Genome Project. This fieldwork was conducted at the experiment station in Hainan in 2019. The experiment was conducted in a growth chamber under 12 h light/12 h dark conditions (26 °C and RH 55%). To screen for the cold stress and chilling acclimation exposure variability in the collected germplasm, shoot fresh weights under different conditions were used for GWAS. For chilling acclimation exposure, cold-stress treatments were performed in the growth chamber at the seedling stage (7-day-old), the rice seedlings were treated by 12 °C (day/night) for 2 days (cold stress), and then, the temperature was decreased to 6 °C (day/night) for 3 days as chilling-acclimation treatment; for cold stress, the seedlings were exposed to 6 °C (day/night) for 3 days then recovered for 9 days. For control, the seedlings continued to grow at 25 °C (day/night). Then, the shoot fresh weight was measured at same time after cold-stress treatment.

In addition, the calculation method of relative value referred to a previous study that used the relative changes in tiller numbers as the calculation of nitrogen response [9]. The relative shoot fresh weight under cold stress was calculated with the value of (shoot fresh weight under normal temperature—shoot fresh weight under cold stress)/shoot fresh weight under normal temperature, and this was used as an effective measurement of the response to cold stress. Similarly, the relative shoot fresh weight under chilling acclimation was calculated with the value of (shoot fresh weight under normal temperature—shoot fresh weight under chilling acclimation)/shoot fresh weight under normal temperature.

### 4.2. Genotyping for GWAS

Genetic variations of 338 accessions were downloaded from https://s3.amazonaws.com/3kricegenome/snpseek-dl/, accessed on 22 June 2022. SNPs were filtered with minor allele frequencies 0.05, missingness per marker 0.02, and missingness per individual 0.01. PLINK was used to deal with the raw data, the principle components (PCs), and kinship matrix analyses. The first two matrices of principle component analysis were used to construct the PC matrix. The LD of this panel was calculated by LD decay software (Version 3.41) [23]. The -log10(p) = 5 was used, and the LD heatmaps were constructed using LDBlockShow software (Version 1.40). The GWAS was performed using the FarmCPU model by rMVP [26]. ANNOVAR was used as to annotate the SNPs [49]. R and TBtools were used as data management and statistical analysis [50].

### 4.3. Integrated Analyses of GWAS and RNA-seq

To reduce the influence of negative positive SNPs, regions with two or more SNPs were selected, and then, regions around these SNPs were considered as QTLs. To find the candidate genes in the QTL, we performed seven RNA-seq (CK, 6 °C 6 h, 6 °C 24 h, 12 °C 6 h, 12 °C 24 h, 12 °C 24 h + 6 °C 6 h, and 12 °C 24 h + 6 °C 24 h) for the cold stresses and chilling acclimation in rice seedlings of Nipponbare (NPB). The sequencing was carried out by BGI Illumina, and the detailed method of RNA-seq analyses and results can be found in our previous study [51]. Then, we integrated genes in QTLs with RNA-seq data, and genes with a higher expression were chosen for further study.

### 4.4. qRT-PCR Analysis of Two Candidate Genes

Seeds of rice variety Zhonghua 11 (ZH11) were surface-sterilized in 70% ethanol for 2 min followed by gentle shaking in the 2.5% NaClO for 30 min and then washed several times with sterilized water. Sterilized seeds were soaked in distilled water for 4 d and germinated in the incubator. After that, seedlings were grown hydroponically in a growth chamber under 12 h light/12 h dark condition at 25 °C for 14 d, with constant illumination intensity of about 500 μmol.m^−2^.s^−1^ and a relative humidity of 55%. Rice seedlings were then subjected to the different chilling stresses, and shoot tissues from six plants were collected at the designated time points, immediately frozen in liquid nitrogen, and stored at −80 °C until use. Total RNA was extracted from leaves of 2-week-old ZS11, and the primers for two candidate genes used are listed below: *LOC_Os01g62410*F: 5′-GACAAGCGGGCCAATAAGGA-3′, *LOC_Os01g62410*R: 5′-ATCGATGCAAGCATTGTACCTCT-3′; *LOC_Os12g24490*F: 5′-TCGTCGTGCTCTACCTGTT-3′, *LOC_Os12g24490*R: 5′-TACTCGAACGCCGGGAT-3′.

## 5. Conclusion

In this study, we focused on novel cold stress and chilling acclimation of rice seedlings. A total of 338 accessions of rice core germplasms was used, and relative shoot fresh weights at different temperature conditions were calculated for GWAS, and candidate genes within 12 QTLs were integrated with RNA-seq. Then, haplotype analysis was carried out for three selected genes. In summary, there is a shared pathway between cold stress and chilling acclimation, but most of the pathways were unique for chilling acclimation or cold stress. This study provides a platform to further analysis of chilling acclimation exposure in rice.

## Figures and Tables

**Figure 1 ijms-23-13208-f001:**
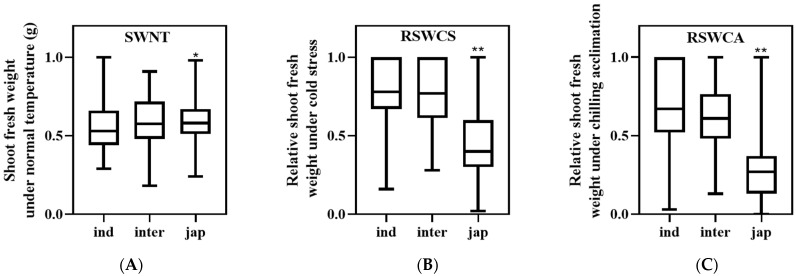
Comparison of shoot fresh weight of rice subpopulations under different temperature conditions. (**A**–**C**) The box plot of the response of three rice subpopulations seedlings to the normal temperature (**A**), cold stress (**B**), and chilling acclimation (**C**). SWNT (g), shoot fresh weight under normal temperature; RSWCS, relative fresh weight after cold stress; RSWCA, relative fresh weight after chilling acclimation; ind, *indica* subpopulation; inter, *intermediate* subpopulation; jap, *japonica* subpopulation. * and ** represent significant differences at the 0.05 and 0.01 levels; respectively.

**Figure 2 ijms-23-13208-f002:**
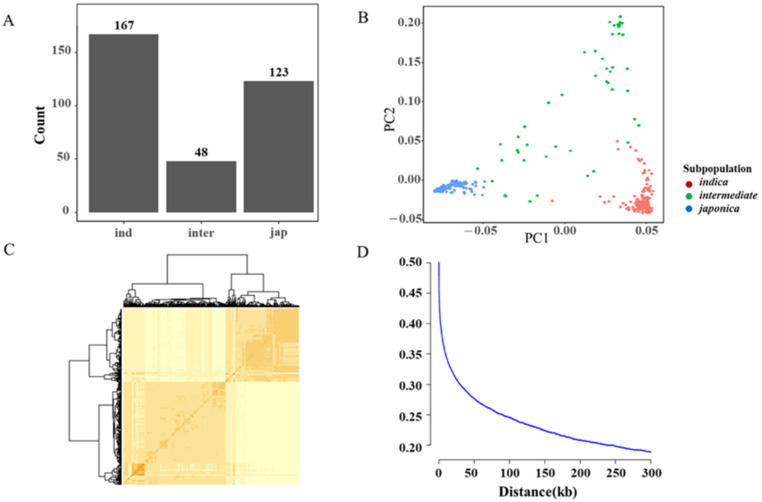
Population genetic analyses and polymorphic SNPs. (**A**) Distribution of 338 rice accessions. (**B**) Principal component analysis of the association panel analyzed by PLINK [25]. (**C**) Kinship analysis of the association panel. (**D**) LD decay calculated by PopLDdecay.

**Figure 3 ijms-23-13208-f003:**
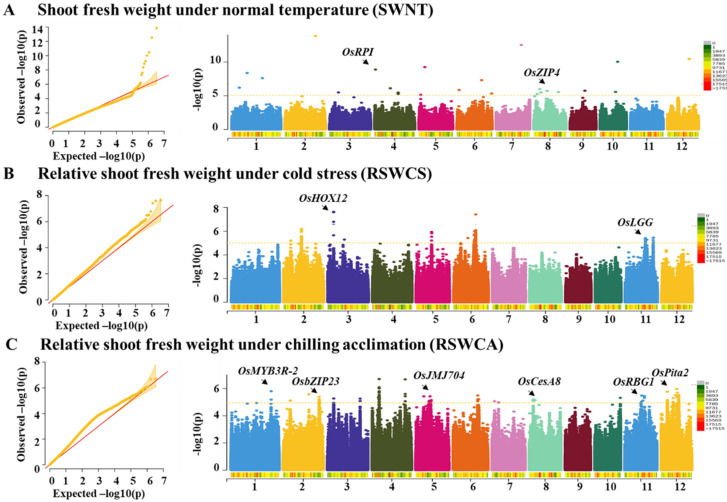
GWAS of the shoot fresh weight under three different conditions. (**A**–**C**) Manhattan plot of shoot fresh weight under normal temperature (SWNT (g)) (**A**), relative shoot fresh weight under cold stress (RSWCS)(**B**), and relative shoot weight under chilling acclimation (RSWCA)(**C**). The arrow is linked to significant SNPs that have been reported.

**Figure 4 ijms-23-13208-f004:**
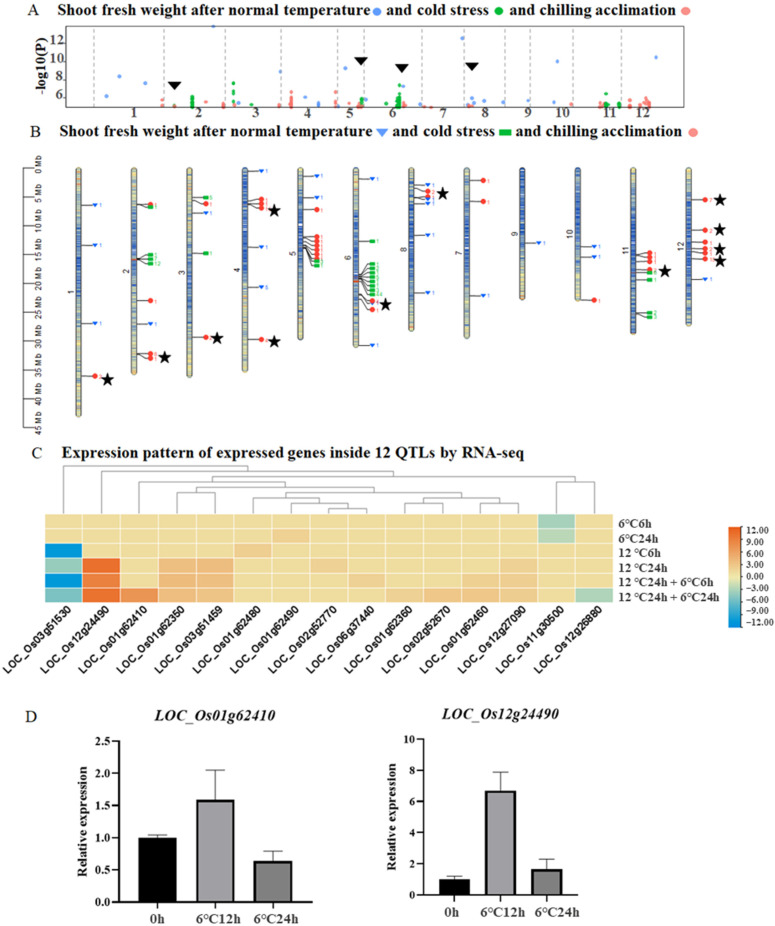
Integrated analysis of SNPs of the shoot fresh weight and expression pattern of genes in 12 QTLs. (**A**) Comparison of the QTL results under three different temperature conditions. (**B**) Distribution of SNPs on different chromosomes. Stars represent QTLs with two or more SNPs and that were specifically detected under chilling acclimation. (**C**) Expression pattern of 15 genes of 12 QTLs. (**D**) Relative expression of *LOC_Os01g62410* and *LOC_Os12g24490* by RT-qPCR.

**Figure 5 ijms-23-13208-f005:**
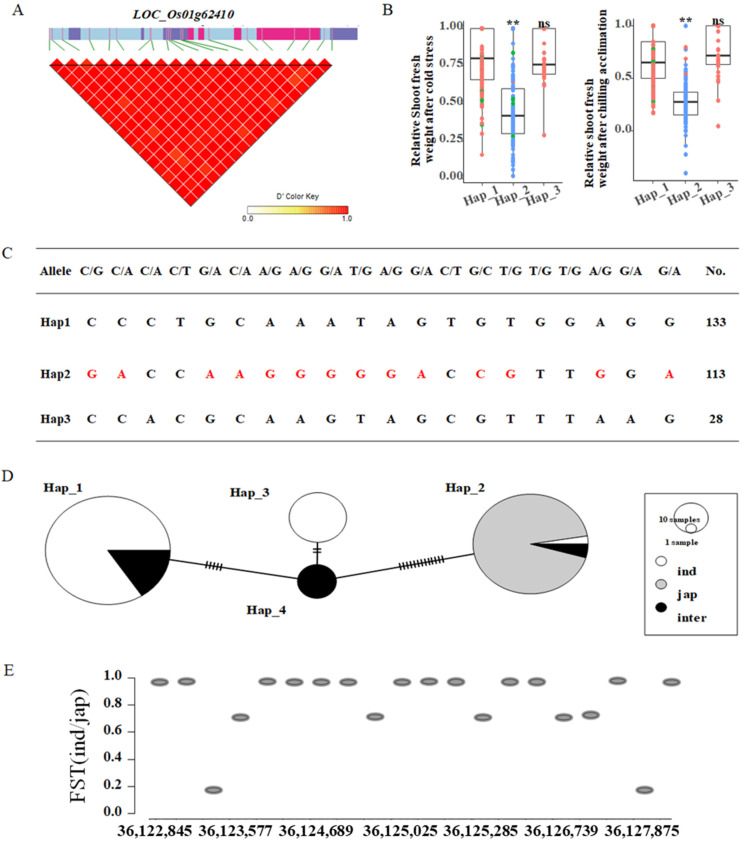
LD block, haplotype, and distribution analyses of the SNPs. (**A**) LD block analysis of *LOC_Os01g62410*. (**B**) Haplotype analysis of *LOC_Os01g62410* under cold stress and chilling acclimation. (**C**) SNPs information of each haplotype. (**D**) Haplotype network of *LOC_Os01g62410* and color represents different rice subpopulations. White represents *indica* accessions, gray represents *japonica* accessions, and black represents *intermediate* accessions. (**E**) FST analysis of *LOC_Os01g62410* between *indica* and *japonica* subpopulation. FST was calculated between *indica* and *japonica* accessions. A, color represents different rice subpopulations. Red represents *indica* accessions, blue represents *japonica* accessions, and the green represents intermediate accessions. ** represent significant differences at 0.01 levels.

**Figure 6 ijms-23-13208-f006:**
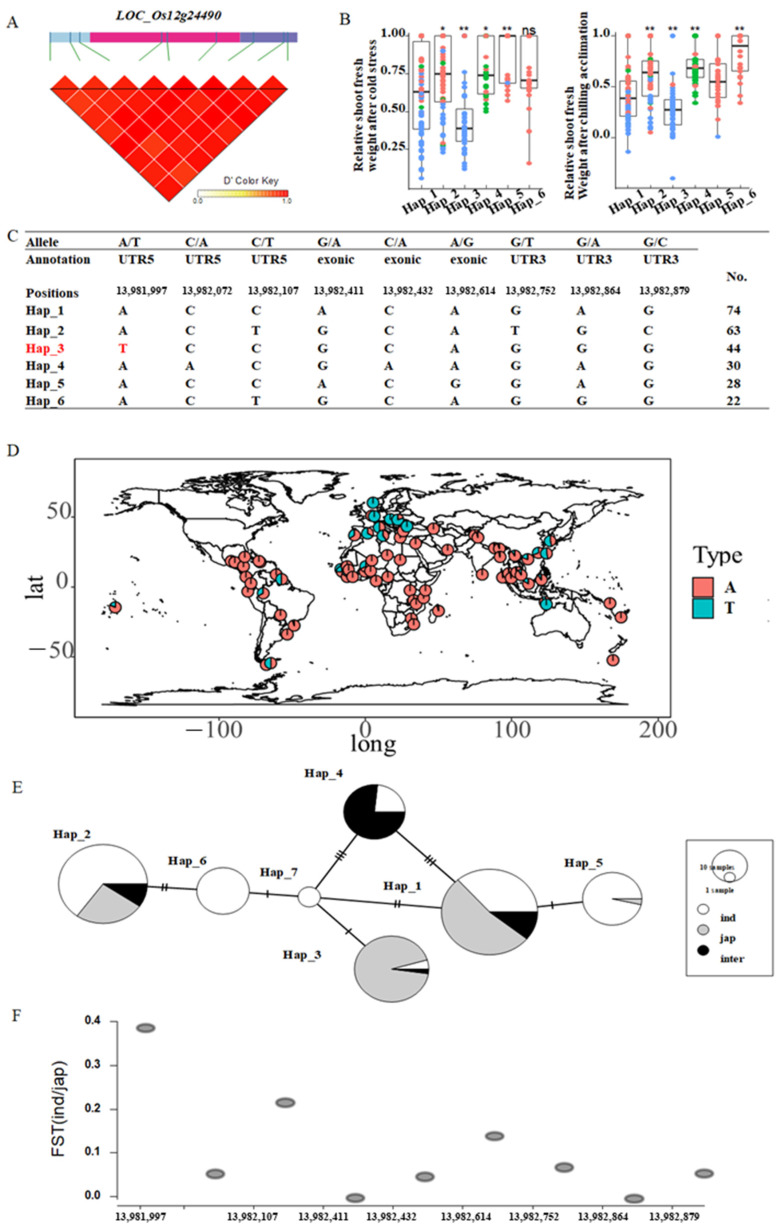
LD block, haplotype, distribution, and FST analyses of the gene *LOC_Os12g24490*. (**A**) LD block analysis of the gene *LOC_Os12g24490*; (**B**) Haplotype analysis of *LOC_Os12g24490* under cold stress and chilling acclimation, (**C**) SNPs information of each haplotype. (**D**) Distribution of the SNP in the gene *LOC_Os12g24490*. A, color represents different rice accessions. Red represents *indica* accessions, blue represents *japonica* accessions, and the green represents *intermediate* accessions. (**E**): Haplotype network of the gene *LOC_Os12g24490,* and color represents different rice subpopulations. White represents *indica* accessions, gray represents *japonica* accessions, and black represents *intermediate* accessions. (**F**): FST analysis of the gene *LOC_Os12g24490* between *indica* and *japonica* subpopulations. FST was calculated between *indica* and *japonica* accessions. * and ** represent significant differences at the 0.05 and 0.01 levels; respectively.

## Data Availability

The information of the 338 accessions of this study are openly available at https://snp-seek.irri.org, accessed on 22 June 2022.

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
