# Peer review of "Genome-Wide Association Mapping Identifies New Candidate Genes for Cold Stress and Chilling Acclimation at Seedling Stage in Rice (Oryza sativa L.)"

_ijms, 2022, doi:10.3390/ijms232113208_

Round 1
Reviewer 1 Report
Comments
Line 76: Please decipher “FarmCPU”.
Line 120: Please decipher “PCA”.
Line 122: “LD of this panel was calculated by LDdecay software” It shut be moved to Materials and Methods.
Line 181-182: Please write why 2 of 3 genes were taken.
Chapter 2.4: It may be worth considering genes in which expression is significantly reduced.
Line 210: Please decipher “FST”.
In the discussion, it is written that the gene LOC_Os12g22490 encodes C3HC4-type RING zinc. It is also necessary to specify what the LOC_Os01g62410 gene encodes, as well as the OsCycB1;1, OsCycB2;1, OsCycB2;2, OsCDC20.1 genes specified in lines 260-261.
In the title you specify "New Genetic Mechanism for Cold Stress and Schilling Acclimation" So what are these mechanisms?
In the discussion, indicate the same and different results compared to other works investigating QTL cold stress and chilling acclimation.
The chapter materials and methods do not specify how the RNA-seq was performed.
Author Response
Comments1
Line 76: Please decipher “FarmCPU”.
Thank you very much for your good suggestion. We had added the full name of abbreviation FarmCPU (Fixed and random model circulating probability unification) in the paper. (Line 83).
Line 120: Please decipher “PCA”.
Thanks. We had added the full name of abbreviation PCA (Principal Component Analysis) in the paper. (Line 132).
Line 122: “LD of this panel was calculated by LDdecay software” It shut be moved to Materials and Methods.
Thanks. We had moved this part to the Materials and Methods. (Line 404).
Line 181-182: Please write why 2 of 3 genes were taken.
Thank you very much for your careful idea. 12 QTLs were found to be involved in chilling acclimation. To find the candidate genes, we investigated their expression under cold stress and found that 15 genes were obviously expressed. Three of them had much significant changes in the expression levels, and two of them are up-regulated in cold stress and chilling acclimation. So, we selected and highlighted these two genes for further analysis. We have added this information in the MS.
Chapter 2.4: It may be worth considering genes in which expression is significantly reduced.
Thank you very much for your careful idea. On the one hand, the expression of LOC_Os03g51530 is similar in cold stress and chilling acclimation. On the other hand, the down-regulated gene LOC_Os03g51530 has not yet annotated. Therefore, we choose these two significantly up-regulated genes for further analysis.
Line 210: Please decipher “FST”.
Thanks. We had added the full name of “FST” (Wright's fixation index), a measure of genetic differentiation between two populations in the paper. (Line 27).
In the discussion, it is written that the gene LOC_Os12g22490 encodes C3HC4-type RING zinc. It is also necessary to specify what the LOC_Os01g62410 gene encodes, as well as the OsCycB1;1, OsCycB2;1, OsCycB2;2, OsCDC20.1 genes specified in lines 260-261.
Thanks. We added the annotation (LOC_Os01g62410 that encodes a R1R2R3 MYB transcription activator) of LOC_Os01g62410 in the paper (line 25).
In addition, we specified the genes as below OsCycB1;1 (LOC_Os01g59120), OsCycB2;1 (LOC_Os04g47580), OsCycB2;2 (LOC_Os06g51110), OsCDC20.1 (LOC_Os02g47180). (Line 281).
In the title you specify "New Genetic Mechanism for Cold Stress and chilling Acclimation" So what are these mechanisms?
Thank you very much for your good suggestion. We had changed the "New Genetic Mechanism for Cold Stress and chilling Acclimation" into “New Candidate Genes for Cold Stress and Chilling Acclimation” (Line 2).
In the discussion, indicate the same and different results compared to other works investigating QTL cold stress and chilling acclimation.
Thank you very much for your good suggestion. We added the comparison of the QTLs in this study with those in some previous studies in Discussion Section (Line 291- 304).
The chapter materials and methods do not specify how the RNA-seq was performed.
We had added the information of RNA-seq in the revised manuscript and cited one of our previous studies in the Materials and Methods section.
Reviewer 2 Report
The work is an important attempt to characterize genes involved in cold acclimatization, however many points are unclear from the writing. In particular, the treatments, the phenological state of the seedlings or the cultivation time are not precise. As the treatments have different times, it is unclear when fresh weight was measured.
On the other hand, fresh weight data at stressful vs normal temperatures are meaningless, especially for Indicas and intermediates, since the values in the supplemental table are higher in the cold than normal temperatures.
line 92: are the data presented in tableS3 are correct? indica accessions and intermediate grew better under cold and chilling acclimation stress than their control (the treatment times differ in cold stress).
In Figure 1: relative data are not shown in % in M&M is cited in the bibliography that the authors used to calculate relative growth, but are is not clear if the calculation formula should be included in M&M.
in line 120: in materials and methods, the authors claim that PC was done by two principal components, but if three genotype groups were used, Why were only two principal components used for the population structure?
in line 127: the marker map in chromosome 10 is the densest, and the Chr 4 is the least dense, and data are wrong according to table S4
in line 143: data showed SNP isolates with very different significance; this could be a case of false positives. While for RSWCS, peak grouping in Chromosomes 2, 5 and 6 and peak grouping in Chromosomes 4 and 6 by RSWCA are not taken into account.
supplementary table S5: legend says, "relative shoot fresh weight under normal temperature". It is at normal temperature; what is relative?
line 315: age of plants?
Author Response
Comments2
The work is an important attempt to characterize genes involved in cold acclimatization, however many points are unclear from the writing. In particular, the treatments, the phenological state of the seedlings or the cultivation time are not precise. As the treatments have different times, it is unclear when fresh weight was measured.
Thank you very much for your critical comments. The statement has been revised as followed. “For chilling acclimation exposure, cold stress treatments in the growth chamber at the seedling stage (7-day-old), the rice seedlings were treated by 12℃ (day/night) for 2 days (cold stress) and then decreased the temperature to 6℃ (day/night) for 3 days as chilling acclimation treatment, for cold stress the seedlings exposure to 6℃ (day/night) for 3 days then recovered for 9 days. For control, the seedlings continue to grow at 25℃ (day/night) for 6 d. Then, the shoot fresh weight was measured at same time after cold stress treatment.” We added the related information in line 361-367.
On the other hand, fresh weight data at stressful vs normal temperatures are meaningless, especially for Indicas and intermediates, since the values in the supplemental table are higher in the cold than normal temperatures.
Thank you very much for your critical comments and good suggestions.
We have added the treatment conditions and cultivation time (7-day-old seedlings) in the MS, and added that the relative shoot fresh weight at cold stress and chilling acclimation was calculated with the value of (shoot fresh weight under normal temperature – shoot fresh weight under cold stress or chilling acclimation)/shoot fresh weight under normal temperature, which is similar to calculation of nitrogen response in Liu’s method (Liu, Y., Wang, H., Jiang, Z. et al. Genomic basis of geographical adaptation to soil nitrogen in rice. Nature 590, 600–605 (2021). https://doi.org/10.1038/s41586-020-03091-w).In this way, we could evaluate the cold response and chilling acclimation response by the relative shoot fresh weight values. The more change, the less cold stress response or less chilling acclimation response in rice seedlings.
line 92: are the data presented in tableS3 are correct? indica accessions and intermediate grew better under cold and chilling acclimation stress than their control (the treatment times differ in cold stress).
Thank you very much for your carefulness. According to the measurement formula of relative shoot fresh weight under cold stress and chilling acclimation. This result is easily to understand that the SWNT (g) is the actual value under normal temperature, and the RSWCS and RSWCA is the relative value under cold stress and chilling acclimation, so the relative value of accessions and intermediate accessions seems bigger than the control under normal temperature conditions, indicating that indica accessions and intermediate accessions were more sensitive to cold stress and chilling acclimation conditions than the japonica accessions.
In Figure 1: relative data are not shown in % in M&M is cited in the bibliography that the authors used to calculate relative growth, but are is not clear if the calculation formula should be included in M&M.
Thanks. We have corrected the error and redraw Figure 1.
in line 120: in materials and methods, the authors claim that PC was done by two principal components, but if three genotype groups were used, Why were only two principal components used for the population structure?
Thanks. In genome-wide association study, the principal components were used to reduce the influence of population. As the figure 2B showed that, the first two principal components were enough to distinguish three types of rice subpopulations. Thus, we just used the two principal components (the first and the second in the rank).
in line 127: the marker map in chromosome 10 is the densest, and the Chr 4 is the least dense, and data are wrong according to table S4
Thanks. The error has been corrected in Line 140-141.
in line 143: data showed SNP isolates with very different significance; this could be a case of false positives. While for RSWCS, peak grouping in Chromosomes 2, 5 and 6 and peak grouping in Chromosomes 4 and 6 by RSWCA are not taken into account.grouping in Chromosomes 4 and 6 by RSWCA are not taken into account.
Thank you very much for your good idea. Peak grouping in chromosomes 2, 5, 6 for RSWCS and peak grouping in chromosomes 4 and 6 for RSWCA were also important QTLs and these QTLs were listed in Table S8. To find the candidate genes, we integrated the genes located in these QTLs with RNA-seq results. In this study, we prefer to explore the significant candidate genes with the altered expression levels in cold stress. However, the other peak groupings are also important and will also be investigated in the future.
supplementary table S5: legend says, "relative shoot fresh weight under normal temperature". It is at normal temperature; what is relative?
You are right. We have written it as “shoot fresh weight under normal temperature".
line 315: age of plants?
Thanks. The statement had been changed to “For chilling acclimation exposure, cold stress treatments in the growth chamber at the seedling stage (7-day-old).” in Line 361.
Round 2
Reviewer 1 Report
The authors have made the necessary changes and the article can be considered for publication.
Reviewer 2 Report
the authors have made the corrections and clarifications suggested previously